# Adaptive Cruise Control Based on Safe Deep Reinforcement Learning

**DOI:** 10.3390/s24082657

**Published:** 2024-04-22

**Authors:** Rui Zhao, Kui Wang, Wenbo Che, Yun Li, Yuze Fan, Fei Gao

**Affiliations:** 1College of Automotive Engineering, Jilin University, Changchun 130025, China; rzhao@jlu.edu.cn (R.Z.); chewb1519@mails.jlu.edu.cn (W.C.); fanyz23@mails.jlu.edu.cn (Y.F.); 2School of Mechanical Engineering, Beijing Institute of Technology, Beijing 100081, China; 3120230321@bit.edu.cn; 3Graduate School of Information and Science Technology, The University of Tokyo, Tokyo 113-8654, Japan; li-yun@g.ecc.u-tokyo.ac.jp; 4State Key Laboratory of Automotive Simulation and Control, Jilin University, Changhun 130025, China

**Keywords:** autonomous driving, adaptive cruise control, safety aware, deep reinforcement learning, projected constrained policy optimization

## Abstract

Adaptive cruise control (ACC) enables efficient, safe, and intelligent vehicle control by autonomously adjusting speed and ensuring a safe following distance from the vehicle in front. This paper proposes a novel adaptive cruise system, namely the Safety-First Reinforcement Learning Adaptive Cruise Control (SFRL-ACC). This system aims to leverage the model-free nature and high real-time inference efficiency of Deep Reinforcement Learning (DRL) to overcome the challenges of modeling difficulties and lower computational efficiency faced by current optimization control-based ACC methods while simultaneously maintaining safety advantages and optimizing ride comfort. Firstly, we transform the ACC problem into a safe DRL formulation Constrained Markov Decision Process (CMDP) by carefully designing state, action, reward, and cost functions. Subsequently, we propose the Projected Constrained Policy Optimization (PCPO)-based ACC Algorithm SFRL-ACC, which is specifically tailored to solve the CMDP problem. PCPO incorporates safety constraints that further restrict the trust region formed by the Kullback–Leibler (KL) divergence, facilitating DRL policy updates that maximize performance while keeping safety costs within their limit bounds. Finally, we train an SFRL-ACC policy and compare its computation time, traffic efficiency, ride comfort, and safety with state-of-the-art MPC-based ACC control methods. The experimental results prove the superiority of the proposed method in the aforementioned performance aspects.

## 1. Introduction

The adaptive cruise control (ACC) system is an advanced driver-assistance system (ADAS) that enables a vehicle to automatically maintain a desired headway distance from the vehicle ahead. This system monitors the speed and distance of the preceding vehicle through various sensing technologies such as radar, LiDAR, and cameras. It then adjusts the control of the vehicle’s speed accordingly to maintain a safe and stable following distance [1]. Recognized as a critical component of autonomous vehicles [2], recent research has demonstrated that this system can also enhance traffic flow by adaptively adjusting the gap between vehicles in response to dynamically changing traffic conditions [3,4], and this viewpoint is further supported by studies [5,6,7]. Currently, ACC primarily encompasses three control methods: classical control theory, Model Predictive Control (MPC), and Reinforcement Learning (RL)-based control.

For simple controllers and plant models, static control theory methods can effectively meet expected control behaviors [8,9]. Cruise control is typically implemented as a simple P, PI, or PID controller [10]. Design methods such as pole placement design are instrumental in achieving desired control characteristics [10]. However, for more complex systems, the optimal tuning of controller gains is challenging, and the algorithms also suffer from lower performance ceilings.

Compared to traditional control theory, MPC offers improved control performance, and ACC technology based on MPC has made significant progress in recent years [11,12,13]. Starting from [14], MPC-based methods have become the dominant approach because MPC optimizes a multi-objective cost function, including fuel economy, driver comfort, and safety during driving. Stanger et al. [15] created additional constraints for MPC from map data for predictive speed control, enhancing the safety of the algorithm. Moser et al. [16] introduced an ACC algorithm within the MPC framework that considers fuel economy, comfort, and safety simultaneously, offering substantial improvements in fuel efficiency and comfort compared to traditional algorithms. In [17], a specific MPC was designed, and a system tuning method was proposed, allowing the desired performance to be adjusted simply by changing two parameters: one for safety and one for comfort. Generally, considering the nonlinearity of vehicle dynamics, various environmental and driving conditions and adjusting the controller’s weights can improve the performance of the system. Refs. [18,19] parameterized the weights used in the MPC cost function as functions of inter-vehicle distance and speed, achieving good experimental results. Nie et al. using MPC enhances fuel efficiency, safety, and comfort in car-following scenarios, achieving up to 13% better fuel economy than traditional PID-based systems in simulations [20]. Zhang et al. proposed a collision-considerate MPC control strategy that significantly enhances the safety of the ACC algorithm [21]. However, MPC requires a model, and computational costs limit the complexity of the model, its ability to handle uncertainties [22,23], and the length of the prediction horizon. Furthermore, ACC based on MPC necessitates accurate vehicle and dynamics models, and errors in these models can impact the effectiveness of actual control. In the real world, obtaining accurate vehicle and dynamics models is often challenging. Wang et al. proposed a data-driven five-model approach that mitigates the issue of model errors to some extent, but it requires a large amount of labeled real data for its implementation [24].

The emergence of deep reinforcement learning (DRL) shows promise in addressing these issues [25,26,27,28,29,30,31,32]. In DRL, agents interact with the environment in a time-discrete manner, receiving rewards at each time step based on their actions and the state of the environment with the objective being to maximize cumulative and discounted rewards. DRL algorithms employ neural networks as function approximators, and those with continuous action spaces are particularly relevant for solving continuous control tasks, especially in autonomous driving and driver assistance systems [33,34]. Once the policy is trained, control commands are directly mapped by the neural network, significantly reducing computational demands for control strategies and addressing the limitation of computational power affecting the prediction range in MPC methods. A key feature of reinforcement learning is self-learning; unlike supervised learning, the learning algorithm does not rely on a labeled dataset. Instead, optimal behaviors are defined solely by the abstract goal of maximizing collected rewards. Notably, DRL algorithms are model-free, implicitly learning from experience, and solving the challenging modeling issues associated with MPC methods.

In recent years, DRL has progressively been explored for application in the lateral and longitudinal control of autonomous driving. Das et al. [35] propose a dynamic adaptive cruise control system based on DRL that effectively adjusts the headway in response to dynamic traffic conditions on freeways and ramps to optimize traffic flow. However, the experiments did not evaluate the safety performance of the policy. After that, Das et al. [36] designed a separate RL agent that identifies and adapts to the optimal Time-to-Collision (TTC) threshold based on rich traffic information gathered from the surrounding environment, including macro- and micro-traffic data. This RL agent interacts with the main RL agent by providing the optimal TTC threshold as feedback to achieve the best inter-vehicle spacing, thereby maximizing traffic flow, driving safety, and comfort. However, the experimental scenarios for this method are extremely simplistic and still exhibit a 1% collision rate, which is significantly higher than the highest functional safety standards for autonomous driving. DRL policies optimized purely for reward maximization are rarely suitable for safety-critical autonomous driving applications. The primary reason why current RL applications in ACC struggle to ensure safety performance is mainly due to the singular focus on maximizing a single reward. Achieving a balance between safety and other performance metrics is challenging, and such methods often fail to ensure consistent safety under unknown traffic conditions. This limitation constrains the application of DRL to simplified ACC simulation scenarios, complicating its deployment in real-world complex traffic environments. Therefore, ACC systems should prioritize safety as a fundamental constraint during the RL policy update process, rather than merely focusing on maximizing rewards.

To address the challenges of modeling complexity and high computational costs associated with MPC-based ACC policies, and the robust safety challenges of ACC with non-safety-aware DRL strategies, this paper proposes a safe DRL-based ACC strategy, namely the Safety-First Reinforcement Learning Adaptive Cruise Control (SFRL-ACC). This method incorporates a safe trust domain into traditional RL approaches, preventing the policy from increasing safe in the pursuit of maximizing rewards. It achieves collision-free adaptive safe cruising with minimal computational power requirements while also offering high levels of comfort and traffic efficiency. During the training process of the ACC policy, the vehicle initially collects trajectory information over a certain time length within a traffic environment. Subsequently, the policy is updated within the safe trust domain, and this process is repeated until the optimal strategy is trained. This work validates the superior performance of the SFRL-ACC strategy in a simulation environment, achieving a zero-collision rate for the first time in DRL-trained adaptive cruise control strategies, and the policy maintains a high consistency with the expected speed when there is no interference from target vehicles.

The main contributions of this work are as follows:

(1) The ACC problem is modeled as a safe DRL formulation Constrained Markov Decision Process (CMDP), wherein a reasonable design of the state space and action space, as well as the reward and safe cost functions, are considered. The state space includes the velocities of the leading and following vehicles and the distance between them, while the action space considers the future desired velocity of the following vehicle. The reward function takes into account safety, the comfort of the following vehicle, and traffic efficiency, whereas the safe cost function focuses solely on the safety of the vehicle. Upon completion of training and deployment, this strategy demonstrates excellent performance.

(2) We propose the Safety-First Reinforcement Learning Adaptive Cruise Control (SFRL-ACC) method based on the Projected Constrained Policy Optimization (PCPO) approach. This method employs three neural networks for decision making, evaluating overall policy performance, and assessing policy safety performance. When updating the policy neural network in reinforcement learning-based adaptive cruise control methods, it first maximizes rewards and then projects the policy into a safe constraint domain, addressing the issue of insufficient safety.

(3) Extensive experiments were carried out across various traffic scenarios, and the results indicate significant enhancements in computational efficiency, ride comfort, and traffic efficiency when compared to control methods based on MPC. The remainder of this paper is organized as follows: Section 2 presents the problem definition and methodological framework, Section 3 provides a detailed introduction to the SFRL-ACC algorithm, including its components and the update process, Section 4 discusses the experimental setup and compares the experimental results and Section 5 summarizes the entire text.

## 2. Problem Definition and Methodological Framework

### 2.1. Problem Definition

In this section, the ACC problem is defined as a typical optimization issue. Initially, the vehicle under control is referred to as the ego vehicle, while the vehicle in the same lane ahead of the ego vehicle is called the preceding vehicle, as illustrated in Figure 1. Defining the velocity of the preceding vehicle as $v_{pre}$ and that of the ego vehicle as $v_{ego}$, with their relative distance being $d_{\mathrm{rel}},$ the Time to Collision (TTC) and the safety distance $d_{s}$ can be calculated as follows:(1)$$TTC=\frac{d_{rel}}{v_{pre}-v_{ego}}$$
(2)$$d_{s}=TTC_{s}\times \left(v_{pre}-v_{ego}\right)$$
where $TTC_{s}$ represents the safe threshold value for TTC. The ACC scenarios are divided into two cases: when the relative distance between the ego vehicle and the preceding vehicle is greater than the safety distance, as shown in the upper part of Figure 1, the control objective is to stabilize the velocity of the ego vehicle around the desired speed $v_{exp};$ when the relative distance is less than the safety distance, as depicted in the lower part of Figure 1, the aim is to control the TTC to exceed the safety threshold by adjusting the anticipated speed of the ego vehicle at the next time step. At the current moment, $v_{pre},v_{ego},$ and $d_{rel}$ can all be obtained in real time through sensors, and the strategy needs to determine the expected velocity adjustment of the ego vehicle for the next time step to achieve the ACC control objectives.

### 2.2. Methodological Framework

The SFRL-ACC methodology introduces a tripartite neural network system comprising a policy network and two value networks focused on rewards and safety costs. This policy network transforms the immediate local states of the self-driving vehicle into a probability distribution for potential actions in the subsequent moment. Concurrently, the reward and safety cost value networks are tasked with assessing the anticipated rewards and safety expenditures under the prevailing policy. The architecture of the SFRL-ACC approach, encompassing both environment interaction sampling and policy evaluation and update, is depicted in Figure 2.

In the SFRL-ACC algorithm, the responsibility of the environment interaction sampling module is to acquire updated neural parameters for the policy and value networks, subsequently utilizing these parameters to sample experiential data from diverse traffic contexts. This module employs a safe DRL formulation, CMDP, to formally express the process of the ego vehicle with safety constraints exploring within the ACC system environment. This exploration culminates in the generation of discrete time-series trajectory data, encompassing states, actions, rewards, and safety costs.

The SFRL-ACC policy evaluation and update module works in tandem with the environment interaction module, utilizing data collected from the environment interaction module to update the policy and value neural networks. It then synchronizes the updated parameters with the environment interaction module for the next cycle of sampling and optimizing, continuing until the desired ACC control performance is achieved. This module employs the PCPO method to solve the CMDP problem, incorporating safety-constrained costs to further restrict the trust region formed by KL divergence. Depending on the current safety level, this method adjusts the rate and direction of policy updates to achieve a safe and efficient policy, aiming to optimize ACC performance while ensuring that constraint costs remain within predefined bounds.

## 3. SFRL-ACC Algorithm

This chapter provides a detailed description of the SFRL-ACC method, which is composed of two parts: Environment Interaction Sampling and Policy Evaluation and Update. Initially, the ACC is modeled as a CMDP problem by defining custom state spaces, action spaces, reward functions, and cost functions. This is followed by an explanation of the update process for the policy neural network, reward value neural network, and risk value neural network after interaction with the environment. The policy neural network is updated using the PCPO algorithm, which involves two steps: maximizing the reward value and then projecting the maximized reward policy onto a risk constraint to coordinate constraint violations. The reward and risk value neural networks are updated using gradient descent based on the distance between predicted values and actual values.

### 3.1. Representation of ACC Problem to a Safe DRL Formulation CMDP

#### 3.1.1. Constrained Markov Decision Process

Markov Decision Processes (MDPs) are widely regarded as a formal framework for expressing the process of an agent navigating through an environment. They inherently represent a discrete-time decision-making architecture. Within the MDP framework, agents participate in games within a global environment with the objective to cooperate or compete to achieve the maximum total expected reward. However, when safety constraints are introduced, the standard MDP framework is insufficient to describe the environment. Therefore, this section introduces the concept of CMDP. This extends Markov games by integrating constraints of safe trust domains, defining a Constrained Markov game as a tuple {S,A,R,C,P,μ}, where

S is the state space, which is composed of the concatenation of local states observed by the agent and optionally, global non-redundant states;*A* represents the set of action space of the agent, where an action $a_{t}\in A$ at the discrete time step *t*;R:S×A×S→R represents the reward function that describes the instant reward from a state $s_{t}$ by taking an action $a_{t}$ to the next state $s_{t+1}$;$C={\left\{C_{i}\right\}}_{i=1,...,N_{c}}$ represents the set of safe functions defined by the specific environment safety constraints (there are $N_{c}$ safe functions), $C_{i}:S_{t}\times A_{t}\times S_{t+1}\to R$ maps the transition tuples to a safe value with thresholds $d_{1},d_{2},...,d_{N_{c}}$;P:S×A×S→[0,1] represents the transition probability distribution from a state $s_{t}\in S$ by taking an action $a_{t}\in A$ to the next state $s_{t+1}\in S$ at the discrete time step *t*;μ:S→[0,1] represents the initial state distribution.

Under the CMDP model, the agent interacts with the environment within discrete time steps. At each time step *t*, each agent generates a state through interaction with the environment and executes an action $a_{t}∼\pi \left(\cdot |s_{t}\right)$ based on its policy π. After executing action $a_{t}$, the agent receives a reward $R(s_{t},a_{t},s_{t+1})$ and its cost $C(s_{t},a_{t},s_{t+1})$. Subsequently, the environment transitions to a new $s_{t+1}∼P\left(\cdot |s_{t},a_{t}\right)$. Upon reaching a terminal state, the agent starts a new episode, beginning from an arbitrary state $s_{0}∼\mu$. When a trajectory $\tau =(s_{0},a_{0},s_{1},...)$ is collected from an epoch, the policy is updated, allowing the agent to continue interacting with the environment using this newly updated policy. The goal of safe DRL is to enable an agent to learn the optimal policy $\pi^{★}$, which, through continual policy updates, maximizes the expected reward return while keeping the safe cost within its constrained range.

Let $J_{C_{i}}\left(\pi \right)$ denote the expected discounted return $J_{C_{i}}\left(\pi \right)$ of policy π with respect to safe cost function $C_{i}$:(3)$$J_{C_{i}}\left(\pi \right)=E_{\tau ∼\pi }\left[\sum_{t}\gamma^{t}C\left(s_{t},a_{t},s_{t+1}\right)\right]$$
where γ∈[0,1) is the discount factor. With the above conditions, the feasible policy set of the CMDP model is
(4)$$\Pi_{C}≜\left\{\forall i,J_{C_{i}}\left(\pi \right)\leq d_{i}\right\}$$
Given that the expected discounted return of policy π with respect to the reward function is $J\left(\pi \right)=E_{\tau ∼\pi }\left[\sum_{t}\gamma_{t}R\left(s_{t},a_{t},s_{t+1}\right)\right]$, the optimal policy $\pi^{★}$ with the largest expectation value of the reward function under the CMDP model is
(5)$$\pi^{★}=\arg\max_{\pi \in \Pi_{C}}J\left(\pi \right)$$

#### 3.1.2. Converting ACC to Safe DRL Model through CMDP

The SFRL-ACC algorithm determines the expected speed of the controlled vehicle in real time; the state space needs to include the speed information $v_{pre}$ of the preceding vehicle, the speed information of the ego vehicle $v_{ego}$, and the distance information $d_{rel}$ between the two vehicles. Therefore, this paper sets the state space of ACC as $S=\left(v_{pre},v_{ego},d_{rel}\right)$. The action space is defined as the expected speed $v_{ego}^{\prime }$ of the ego vehicle at the next time step. Subsequently, the vehicle’s motion control layer can generate the required throttle opening and brake pad force for the vehicle’s longitudinal control.

Following the CMDP framework, we have established not only a reward function that characterizes the degree of overall performance optimization but also a safe cost function that represents the safety performance of the system. This is to ensure that policy updates maximize the overall performance without violating the vehicle’s safe constraints. To better assess the quality of the policy, we have considered both dense and sparse evaluation items. Dense evaluation items are used to assess the performance at each time step, such as the vehicle’s speed and acceleration, whereas sparse evaluation items are used to assess the vehicle’s performance in each episode and are only triggered by certain special events, such as a collision (termination state) or successful arrival at the destination (termination state).

The safe cost function emphasizes traffic safety and the potential for collision avoidance. Therefore, this paper has designed a TTC safety threshold $TTC_{s}$. When the TTC between two vehicles with potential for collision is less than $TTC_{s}$, the value of the cost function increases by $\varepsilon_{d}$. Hence, the dense loss function $c_{d}$ can be defined as follows:(6)$$c_{d}=\sum_{t=1}^{n}\varepsilon_{d}\delta_{d}$$
Here, $\delta_{d}=1$ represents that there is a collision safe between the ego vehicle and preceding vehicle, meaning the TTC between the vehicles is less than the safe threshold, and $\delta_{d}=0$ indicates that there is no collision safe for the ego vehicle. Additionally, if a collision occurs, the value of the safe function will increase by $\varepsilon_{c}$. Therefore, the sparse loss function $c_{s}$ is defined as
(7)$$c_{s}=\varepsilon_{c}\delta_{c}$$
where $\delta_{c}=1$ indicates that the ego vehicle experiences a collision during its attempt in the environment, and $\delta_{c}=0$ indicates that the ego vehicle successfully reaches its destination in the attempt. The overall safe cost function is defined as
(8)$$C_{SFRL-ACC}=c_{d}+c_{s}$$

The reward function focuses on evaluating comprehensive control performance, encompassing three aspects: safety, comfort, and efficiency. The algorithm’s reward function is composed of dense rewards and sparse rewards. The dense reward function $r_{d}$ is defined as
(9)$$r_{d}=\sum_{t=1}^{n}-\;\varepsilon_{v}\left|v_{con}-v_{exp}\right|-\varepsilon_{a}a_{con}$$
where $\varepsilon_{v},\varepsilon_{a}$ represent weights, $v_{exp}$ denotes the expected velocity when the ego vehicle is in ACC, and $a_{exp}$ represents the real-time acceleration of the ego vehicle. When the ego vehicle successfully reaches the target location, the reward function will increase by $\varepsilon_{s}$. The sparse reward function $r_{s}$ is defined as
(10)$$r_{s}=\varepsilon_{s}\delta_{s}$$
where $\delta_{s}=1$ indicates that the ego vehicle has successfully reached the target location; otherwise, it is 0. The algorithm’s reward function is defined as
(11)$$R_{SFRL-ACC}=r_{d}+r_{s}-C_{SFRL-ACC}$$

### 3.2. Evaluation and Updating of the Policy

#### 3.2.1. Policy Neural Network Optimization

The policy optimization algorithm searches for the optimal feasible policy solution to the CMDP problem. It iteratively updates the policy by maximizing the expected discounted reward J(π) over the intersection of the KL divergence trust region and the risk trust region $\Pi_{\theta }\cap \Pi_{C}$:$$\pi_{k+1}=\arg\max_{\pi }E_{s_{0}∼\rho^{0},a_{0:\infty }∼\pi_{k},s_{1:\infty }∼P_{\pi_{k}}}J\left(\pi \right)$$
$$s.t.\;D_{KL}\left(\pi |\right|\pi_{k})\;\leq \delta$$
(12)$$J_{C_{i}}\left(\pi_{k+1}\right)≜E_{s_{0}∼\rho^{0},a_{0:\infty }∼\pi_{k},s_{1:\infty }∼P_{\pi_{k}}}\left[\sum_{t=0}^{\infty }\gamma^{t}C_{i}\left(s_{t},a_{t}\right)\right]\leq d_{i},\;\forall i=1,...,N_{c}$$
where $N_{C}$ is the number of safe cost functions. Kakade and Langford [37] provide an identity that characterizes the expected safety regulatory value of policy $\pi_{k+1}$ using the safety regulatory advantage function of policy $\pi_{k}$:(13)$$J_{C_{i}}\left(\pi_{k+1}\right)-J_{C_{i}}\left(\pi_{k}\right)=\frac{1}{1-\gamma }E_{s_{0}∼\rho^{0},a_{0:\infty }∼\pi_{k},s_{1:\infty }∼P_{\pi_{k}}}\left[A(s,a)\right]$$
Combining Equations (12) and (13), we can obtain the updatable set of policy $\pi_{k}$ that satisfies the safety regulatory threshold, which is represented as
(14)$$J_{C_{i}}\left(\pi_{k}\right)+\frac{1}{1-\gamma }E_{s_{0}∼\rho^{0},a_{0:\infty }∼\pi_{k},s_{1:\infty }∼P_{\pi_{k}}}\left[A(s,a)\right]\leq d_{i},\;\forall i=1,...,N_{c}$$
Equation (Equation 12) is thus approximated as
$$\pi_{k+1}=\arg\max_{\pi }E_{s_{0}∼\rho^{0},a_{0:\infty }∼\pi_{k},s_{1:\infty }∼P_{\pi_{k}}}\left[J\left(\pi \right)\right]\;$$
$$s.t.\;D_{KL}\left(\pi |\right|\pi_{k})\;\leq \delta$$
(15)$$J_{C_{i}}\left(\pi_{k}\right)+\frac{1}{1+\gamma }E_{s_{0}∼\rho^{0},a_{0:\infty }∼\pi_{k},s_{1:\infty }∼P_{\pi_{k}}}\left[\sum_{t=0}^{\infty }\gamma^{t}C_{i}\left(s_{t},a_{t}\right)\right]\leq d_{i},\;\forall i=1,...,N_{c}$$

The aforementioned formula can increase the expected reward and satisfy specific constraints $d_{i}$. However, for neural networks with high-dimensional parameter spaces, directly solving Equation (Equation 15) may be impractical. Given a small step size δ, the objectives and safety constraints of policy π can be approximated by a linear function around the current policy $\pi_{k}$ and the KL divergence constraint, and $\theta_{k}$ represents the parameters of the policy neural network under policy. Equation (Equation 15) can be approximated by a second-order expansion as follows:$$\theta_{k+1}=\arg\max_{\theta }g^{T}\left(\theta -\theta_{k}\right)$$
(16)$$s.t.\;\frac{1}{2}{\left(\theta -\theta_{k}\right)}^{T}H\left(\theta -\theta_{k}\right)\leq \delta ;\;{\hat{p}}^{T}\left(\theta -\theta_{k}\right)+\hat{c}\leq \delta$$
where $g=\nabla_{\theta }E_{s_{0}∼\rho^{0},a_{0:\infty }∼\pi_{k},s_{1:\infty }∼P_{\pi_{k}}}\left[A_{\pi }\left(s,a\right)\right]$ represents the gradient of the reward advantage function, *H* represents the Hessian matrix of the KL divergence between new and old policies, $\hat{p}=\sum_{i}\nabla_{\theta }J_{C_{i}}\left(\pi \right)$ represents the policy gradient of the safe function, and $\hat{c}=\sum_{i}\sum_{j}\left(J_{C_{i}}\left(\pi \right)-\frac{c_{i}}{1-\gamma }\right),$$\forall i=1,...,N_{c}$ reflects the proximity of the current policy’s safe value to the safety threshold.

During the policy optimization process based on Equation (Equation 16), different levels of safety issues are inevitably encountered. To enhance the efficiency of policy training, the algorithm employs distinct optimization methods for different safe levels. The policy is categorized into three safe levels: low safe, medium safe, and high safe. The algorithm determines the current and post-update safe levels of the policy using three indicators: $\hat{p}$, $\hat{c}$, and *K*. Here, a larger $\hat{p}$ indicates significant changes in safe value after the policy update, a positive value of $\hat{c}$ indicates a higher safe level above the threshold, and a negative value indicates a lower safe level below the threshold. *K* is defined as $K=\delta -\frac{{\hat{c}}^{2}}{\hat{p}H^{-1}\hat{p}}$, where *H* represents the Hessian matrix of the KL divergence between new and old policies. A positive *K* suggests that the KL divergence trust region and the safety regulatory constraint trust region intersect, while a negative *K* indicates that these trust regions are either encompassing or unrelated.

When $\hat{p}$ is extremely small ($\hat{p}\leq 1$ × $10^{-8}$), indicating that the current policy and its neighborhood are in a state of high safety, updating the policy in any direction is safe. When K<0 and $\hat{c}<0$, it implies that the current policy’s KL divergence trust region is within the safe constraint trust region, and updating the policy in any direction is also safe. In these two cases, the algorithm utilizes TRPO [38] to optimize the policy network. It iteratively updates by maximizing the reward advantage function over a local neighborhood of the most recent policy iteration $\theta_{k}$:(17)$$\theta_{k+1}=\arg\max_{\theta }g^{T}\left(\theta -\theta_{k}\right)\;s.t.\;\;\frac{1}{2}{\left(\theta -\theta_{k}\right)}^{T}H\left(\theta -\theta_{k}\right)\leq \delta$$
The network update formula is derived using convex optimization methods to solve Equation (Equation 17):(18)$$\theta_{k+1}=\theta_{k}+\sqrt{\frac{2\delta }{g^{T}H^{-1}g}}H^{-1}g$$

When K>0, indicating that the safe trust region and the KL divergence trust region intersect, maximizing the expected reward value for updates may lead to the policy entering a more dangerous area, representing a medium-safe state. In this case, an additional step is added to the TRPO to ensure that the updated policy satisfies the safe threshold constraint. The update consists of two steps: a reward enhancement step and a projection step. The reward enhancement step maximizes the reward value through the TRPO algorithm, and the projection step projects the policy network after the reward enhancement step into the safe constraint trust region. Initially, the algorithm maximizes the advantage function $A_{\pi }\left(s,a\right)$ within the KL divergence trust region:(19)$$\theta_{k+\frac{1}{2}}=\arg\max_{\theta }g^{T}\left(\theta -\theta_{k}\right)\;s.t.\;\;\frac{1}{2}{\left(\theta -\theta_{k}\right)}^{T}H\left(\theta -\theta_{k}\right)\leq \delta$$

Subsequently, the algorithm projects the intermediate policy $\theta_{k+\frac{1}{2}}$ into the safe constraint trust region by minimizing the distance between $\theta_{k+\frac{1}{2}}$ and the safe constraint trust region:(20)$$\theta_{k+1}=\arg\min_{\theta }\frac{1}{2}{\left(\theta -\theta_{k+\frac{1}{2}}\right)}^{T}H\left(\theta -\theta_{k+\frac{1}{2}}\right)\;s.t.\;\;{\hat{p}}^{T}\left(\theta -\theta_{k}\right)+\hat{c}\leq 0$$

Convex optimization methods are employed to solve Equations (19) and (20).
(21)$$\theta_{k+1}=\theta_{k}+\sqrt{\frac{2\delta }{g^{T}H^{-1}g}}H^{-1}g-\max\left(0,\frac{\sqrt{\frac{2\delta }{g^{T}H^{-1}\hat{p}}}{\hat{p}}^{T}H^{-1}g+\hat{c}}{{\hat{p}}^{T}H^{-1}\hat{p}}\right)H^{-1}\hat{p}$$

If K<0 and $\hat{c}>0$, this indicates that the current policy is in a relatively dangerous state and there is no intersection between the KL divergence trust region and the safety regulatory trust region. In such a situation, updating the policy in any direction will not result in a high-safe state. However, employing the update method for medium-safe states in this scenario can lead to the convex optimization being unsolvable. Therefore, the algorithm alters the method for updating the policy neural network, employing a linear backtracking approach to update the policy neural network with the aim of minimizing the safe value:(22)$$\theta_{k+1}=\theta_{k}-\sqrt{\frac{2\delta }{{\hat{p}}^{T}H^{-1}\hat{p}}}H^{-1}{\hat{p}}^{T}$$

#### 3.2.2. Reward and Cost Value Networks Optimization

For the reward value neural network and the safe value neural network, their gradients are used to update the respective network parameters.
(23)$$\phi_{R}=\arg\min_{\phi }E\left[{\left(V_{\phi_{R}}\left(s_{t}\right)-{\hat{R}}_{t}\right)}^{2}\right]$$
(24)$$\phi_{C}=\arg\min_{\phi }E\left[{\left(V_{\phi_{C}}\left(s_{t}\right)-{\hat{C}}_{t}\right)}^{2}\right]$$

In this context, $\phi_{R}$ and $\phi_{C}$, respectively, represent the parameters of the reward value neural network and the safety value neural network. $V_{\phi_{R}}\left(s_{t}\right)$ and $V_{\phi_{C}}\left(s_{t}\right)$ also denote the evaluated values of the reward and safety values at $s_{t}$, while ${\hat{R}}_{t}$ and ${\hat{C}}_{t}$ correspond to the true values of the reward and safety values, respectively. After updating the three neural networks, Algorithm 1 employs the new policy network to collect trajectories in the environment again, followed by the evaluation and updating of the neural networks, until the policy achieves the desired performance.
**Algorithm 1** SFRL-ACC  1:Initialize $\phi_{R}$, $\phi_{C}$, π, set $d_{C}$, γ, ρ, $N_{e}$, $N_{t}$  2:**for** epoch $k=1,2,\ldots ,N_{e}$ **do**  3:      **for** $t=1,2,\ldots ,N_{t}$ **do**  4:            For ego vehicle, receive state $s_{t}=\left(v_{pre},v_{ego},d_{rel}\right)$, choose an action $a_{t}=v_{ego}^{\prime }$ according to current policy $\pi_{k}$.  5:            Execute action $a_{t}$, get reward $r_{t}$, cost $c_{t}$, next state $s_{t+1}$, w.r.t. current policy and exploration in environment interaction sampling module.  6:            $s_{t}=s_{t+1}$  7:      **end for**  8:      Collect trajectories $\tau =(s_{t},a_{t},r_{t},c_{t},s_{t+1})$  9:      Calculate advantage function of reward and risk function: $A_{r}\left(s,a\right),A_{c}\left(s,a\right)$10:      Calculate $\hat{g},\hat{p},\hat{c},K$:11:      $\hat{g}=\nabla_{\theta }J\left(\tau \right)$, $\hat{p}=\sum_{i}\nabla J_{C_{i}}\left(\tau \right)$, $\hat{c}=\sum_{i}\left(J_{C_{i}}\left(\pi \right)-\frac{c_{i}}{1-\gamma }\right)$, $K=\delta -\frac{{\hat{c}}^{2}}{{\hat{p}}^{T}H^{-1}\hat{p}}$12:      **if** High Safety **then**13:            Update policy network as:14:            $\theta_{k+1}=\theta_{k}-\sqrt{\frac{2\delta }{g^{T}H^{-1}g}}H^{-1}g$15:      **else if** Medium Safety **then**16:            Solve convex dual problem, get $v^{*}$, $\lambda^{*}$17:            Solve α by backtracking line search, update policy network as:18:            $\theta_{k+1}=\theta_{k}+\frac{\alpha }{\lambda^{*}}H^{-1}\left(\hat{g}-\hat{b}v^{*}\right)$19:      **else**20:            Update policy network as:21:            $\theta_{k+1}=\theta_{k}-\sqrt{\frac{2\delta }{{\hat{b}}^{T}H^{-1}\hat{b}}}H^{-1}\hat{b}$22:      **end if**23:      Update $\phi_{R}$, $\phi_{C}$ as:24:      $\phi_{R}=\arg\min_{\phi }E\left[{\left(V_{\phi_{R}}\left(s_{t}\right)-{\hat{R}}_{t}\right)}^{2}\right]$25:      $\phi_{C}=\arg\min_{\phi }E\left[{\left(V_{\phi_{C}}\left(s_{t}\right)-{\hat{C}}_{t}\right)}^{2}\right]$26:**end for**

### 3.3. Algorithm Overview

The SFRL-ACC method is divided into training and deployment phases. During training, it first initializes the parameters of the policy, reward, and safety value neural networks as well as some other necessary experimental parameters (line 1). Subsequently, it engages with the environment using the initialized policy. For the ego vehicle, it receives state space information from the environment, and the policy neural network guides the action space, leading the vehicle into the next state (lines 2–7). After the ego vehicle completes a round of interaction with the environment, the policy neural network is optimized based on the trajectory information collected from this interaction (lines 8–19). When the control policy is at a high safety level, optimization is conducted using the TRPO algorithm (lines 11–12); at a medium safety level, the PCPO algorithm is employed for optimization (lines 13–15); and at a low safety level, a linear backtracking method is used to minimize the safety value (lines 16–17). Finally, the parameters of the reward and safety value neural networks are updated using gradient-based methods to minimize the gap between estimated and true values (line 18). Upon completion of the training, the optimized parameters of the AFRL-ACC strategy neural network are obtained and deployed in a traffic environment. This setup allows for the real-time translation of ACC system state inputs into behavioral outputs, enabling the effective control of vehicles.

## 4. Experiment

### 4.1. Experimental Setting

To evaluate the proposed SFRL-ACC algorithm in experimental tests, this section conducts two sets of experiments. Specifically, the first experiment demonstrates the training process of the SFRL-ACC algorithm, showcasing the evolution of performance during the iterations of policy updates and the final performance of the policy. The second experiment compares the performance of a baseline control strategy based on MPC [20] with the SFRL-ACC policy post-training under various different traffic scenarios. Compared to the MPC method described in the original text, we made minor adjustments. To ensure a fair comparison of performance under the same scenarios, the parameters related to the experimental setup in this paper, including the desired cruising speed and maximum permissible vehicle speed, were adjusted. All other parameters related to the algorithm were kept consistent with the original text.

All experiments were conducted in a simulated environment, where an urban highway scenario was constructed using Carla 0.9.12, and the safe DRL model was built based on the Pytorch framework. Additionally, Carla’s builtin sensors were used for real-time transmission of vehicle state information, and the BasicAgent class was employed to generate vehicular trajectories in the urban highway scenario. The desired vehicle speed output by the policy neural network was converted into throttle and brake control signals through Carla’s built-in PID controller to control the vehicle. The GPU used was NVIDIA GeForce RTX 3090 (NVIDIA, New York, NY, USA), and the operating system was Ubuntu 18.04.

In this experiment, we used Carla TOWN 05’s urban highway scenario to train and test the SFRL-ACC algorithm, providing a simulation environment that balances control and realism for assessing the algorithm’s performance across a range of traffic ACC conditions. This study defines a single trial, in accordance with the international standard ISO 15622 [39], as one complete circuit around the urban highway, which includes scenarios such as constant speed following, preceding vehicle braking, cut in, and randomly occurring cut out. In the testing environment following policy deployment, each condition is triggered randomly. Moreover, the location of each trigger was also random, which closely aligns with real driving scenarios. To approximate the real-world scenarios as closely as possible, we collected the PID parameters of the underlying tracking control from actual vehicles and deployed them into the Carla simulator. The target cruising speed of the ego vehicle was set to 16 m/s with a time step of t = 0.1 s. The architecture for both policy and value neural networks was 3 × 128 × 128 × 1. For each policy iteration, n = 2048 samples were collected, and the reward and safe function neural networks were optimized using the Adam optimizer. The learning rate was linearly decayed from 1 × $10^{-3}$ to 0, and the training algorithm was halted after 2000 iterations of updates. Please refer to Table 1 for detailed experimental parameters.

### 4.2. Analysis of Performance during Training Process

In this experiment, we successfully trained a high-performing control policy using the SFRL-ACC algorithm and analyzed the evolution of reward and safe values during the training process. The training scenario is shown in Figure 3. Figure 4a illustrates the changes in reward values throughout the training process. The curve in the graph represents the mean of the training data, while the shaded area indicates the variance in reward values. The initial state of training is the lowest point in the process, where the control policy is unable to complete the adaptive cruise around the urban highway without collisions, resulting in negative scores. During the first hundred iterations of policy iteration, there is a rapid increase in reward values, indicating occasional successful completions of the route without collisions. Between one hundred and five hundred iterations, the reward values fluctuate significantly around zero. This fluctuation is due to the random occurrence of hazardous scenarios in the training environment, which the policy has not yet learned to handle effectively. From five hundred to twelve hundred iterations, there is a gradual increase in reward values as the policy progressively masters the task in ACC, leading to an increasing number of successful completions of the target route. Eventually, after twelve hundred iterations, the reward values stabilize at a very high score, signifying the policy’s excellent performance in terms of safety, comfort, and traffic efficiency.

Figure 4b displays the evolution of safe cost values during the training process, where the safe is associated only with two factors: adherence to TTC between vehicles and the occurrence of collisions. At the beginning of training, the safe cost value is extremely high, corresponding to the very low reward values, indicating that at this stage, the ego vehicle is almost certain to collide in each attempt. During the first hundred iterations, the safe cost value rapidly decreases, marking occasional successful trials. In the iterations from one hundred to five hundred, the safe cost value fluctuates around four hundred. Between five hundred and twelve hundred iterations, the safe cost value gradually decreases, eventually falling below the safe threshold after twelve hundred iterations. The trend in safe values corresponds with the changes in reward values, signifying that the reward function is strongly related to safety performance. The direction of policy training is oriented toward enhancing performance aspects like comfort and traffic efficiency while ensuring safety.

### 4.3. Performance Comparison Post-Policy Deployment

This section compares the performance of the policy trained using the SFRL-ACC algorithm post-deployment with that of a control policy based on MPC [2]. For performance testing post-policy deployment, three typical scenarios were used: constant speed follow, preceding vehicle braking and cut in. Typically, these scenarios are relatively common and challenging in complex traffic environments, and they are used to verify the comprehensive performance of the algorithm. For each scenario, we conducted tests in six sets, with each set consisting of ten trials, to verify the stability and robustness of the algorithm. Typically, these scenarios are relatively common and challenging in complex traffic environments, and they are used to verify the comprehensive performance of the algorithm.

#### 4.3.1. Constant Speed Follow Scenario

Figure 5 presents a performance comparison between the MPC-based ACC strategy and the SFRL-ACC method in the constant speed follow scenario, focusing on speed tracking, acceleration, and the number of near collisions with the preceding vehicle. As evident from Figure 5a, the SFRL-ACC demonstrates a smoother velocity change process in basic car following, which is attributed to its superior real-time performance. The MPC-based ACC policy, due to its higher computational demands and longer computation times, shows significant latency in velocity adjustment, leading to larger velocity fluctuations. Similarly, under steady car-following conditions, the MPC-based ACC experiences greater acceleration fluctuations, as illustrated in Figure 5b. Figure 5c depicts the TTC safety threshold violations for both methods. In a constant speed follow scenario, both vehicles manage to meet safety requirements, maintaining a reasonable TTC with the preceding vehicle. Figure 5d presents a comparison of the fuel consumption per hundred kilometers for two methods. It is clear from the figure that the mean fuel consumption per hundred kilometers of the SFRL-ACC method is consistently lower than that of the MPC method. This can be attributed to the comfort-related reward function embedded within the SFRL-ACC approach, where increased comfort leads to fewer instances of rapid acceleration and deceleration, thereby conserving energy. Both methods exhibit minor fluctuations with similar magnitudes, which are due to the high level of randomness in the testing environment.

#### 4.3.2. Preceding Vehicle Braking Scenario

The performance comparison with the scenario where the preceding vehicle is braking is illustrated in Figure 6. The three curves in the figure represent the preceding vehicle, the ego vehicle using the MPC method, and the ego vehicle using the SFRL-ACC algorithm, respectively.

Figure 6a shows the speed changes of the ego vehicles during emergency braking by the preceding vehicle. It can be observed that the preceding vehicle is accelerating in the first 60 time steps and suddenly decelerates between time steps 60 and 70, posing a rear-end collision safe to the ego vehicle, which needs to decelerate rapidly to avoid collision. The controlled vehicle based on the SFRL-ACC algorithm reacts almost without delay, beginning to decelerate within two to three time steps of the preceding vehicle’s deceleration. In contrast, the ego vehicle using the MPC method shows a significant delay, primarily due to the longer computational time of the MPC algorithm, which delays the response to the preceding vehicle.

Figure 6b presents the changes in acceleration during the emergency braking scenario. Acceleration is a classic indicator of ride comfort with lower acceleration peaks indicating better comfort. The ego vehicle using the SFRL-ACC algorithm exhibits smaller acceleration peaks, benefiting from the lower delay of the SFRL-ACC algorithm, allowing the vehicle adequate time to decelerate rather than requiring rapid deceleration, thus indicating better ride comfort.

Figure 6c illustrates the TTC safety threshold violations for both algorithms. From the figure, it is evident that the MPC-based ACC algorithm experiences a few instances of TTC safety threshold violations, indicating certain moments of danger, which are attributed to the modeling errors and higher computational demands of the MPC method. On the other hand, the SFRL-ACC algorithm consistently remains within the safety range throughout, demonstrating the robustness and safety of the SFRL-ACC algorithm. Figure 6d displays a comparison of the per hundred-kilometer fuel consumption between the two methods, highlighting the superior performance of the SFRL-ACC.

#### 4.3.3. Cut-In Scenario

Figure 7 demonstrates the performance validation in a cut-in scenario. In the figure, the vehicle accelerates normally for the first fifty time steps, and between 50 and 60 time steps, the preceding vehicle cuts in from a neighboring lane in front of the target vehicle and then continues to travel in front of it. Figure 7a shows the velocity change process; it can be seen that when the lead vehicle starts to change lanes, the control vehicle based on the SFRL-ACC algorithm initially decelerates to give way, whereas the target vehicle based on MPC begins to decelerate to give way after a delay of dozens of time steps, reflecting the lower computational power required by the neural network and the superiority of the safety RL method. Figure 7b presents the acceleration change process, validating the velocity change process, where the control method based on MPC exhibits greater latency and larger acceleration peaks, indicating a poorer riding experience. As for Figure 7c, it illustrates the TTC safety threshold violations for both methods. The MPC method, due to its higher computational demands and modeling errors, experienced several instances where it fell below the TTC safety threshold during the control process, indicating a risk of collision. Moreover, compared to the preceding vehicle braking scenario, there were more moments of violation, which were attributed to the higher danger inherent in the cut-in scenario. In contrast, the control vehicle based on the SFRL-ACC algorithm consistently maintained a safe distance from the threshold, thereby demonstrating the superiority of the SFRL-ACC algorithm. Figure 7d displays a comparison of the per hundred-kilometer fuel consumption between the two methods, highlighting the superior performance of the SFRL-ACC.

## 5. Conclusions

This paper introduces a new adaptive cruise control system, named the SFRL-ACC, which leverages the model-free and high real-time inference efficiency of DRL to address the challenges in modeling and computational efficiency faced by current optimization control-based ACC methods. The SFRL-ACC system not only maintains the safety advantages but also optimizes ride comfort. To achieve this, the ACC problem is transformed into a safe DRL formulation, CMDP, through the careful design of state, action, reward, and cost functions. A key component of this system is the PCPO algorithm, which is specifically developed for solving the CMDP problem. PCPO integrates safety constraints into the DRL policy updates by restricting the trust region defined by the KL divergence, thus ensuring performance maximization within safe limits. The paper concludes with a comparative analysis of the SFRL-ACC policy against current state-of-the-art MPC-based ACC methods, demonstrating its superiority in computation time, traffic efficiency, ride comfort, and safety. This establishes the SFRL-ACC as a significant advancement in the field of autonomous vehicle control.

## Figures and Tables

**Figure 1 sensors-24-02657-f001:**
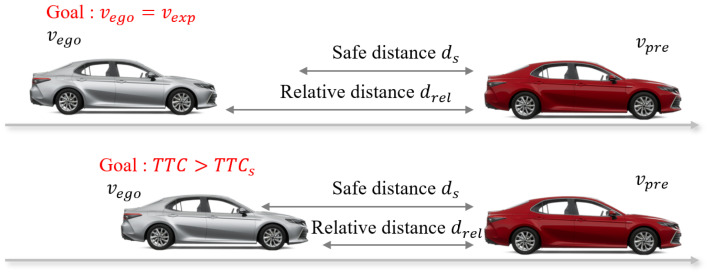
Schematic for two-car following.

**Figure 2 sensors-24-02657-f002:**
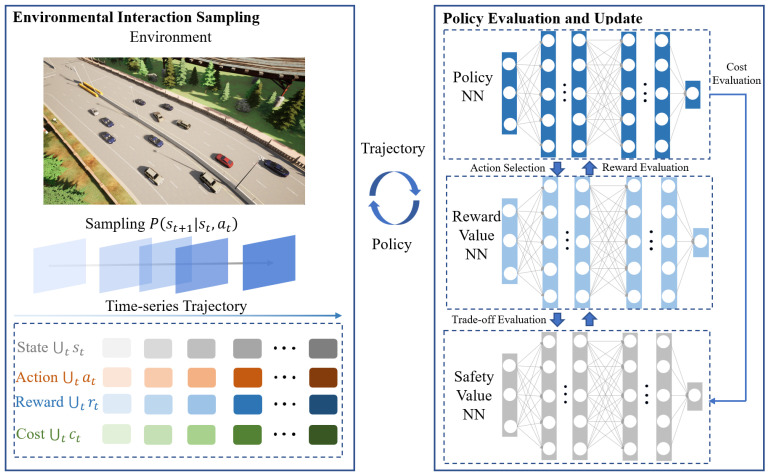
Overall framework diagram of the SFRL-ACC method divided into two parts: Environment Interaction Sampling and Policy Evaluation and Update.

**Figure 3 sensors-24-02657-f003:**
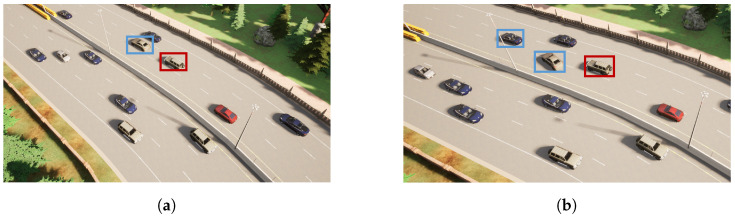
The training scenarios of the SFRL-ACC algorithm. The blue box represents the preceding vehicle, and the red box represents the ego vehicle. Scenario (**a**) represents preceding vehicle braking, while scenario (**b**) signifies a vehicle cut in.

**Figure 4 sensors-24-02657-f004:**
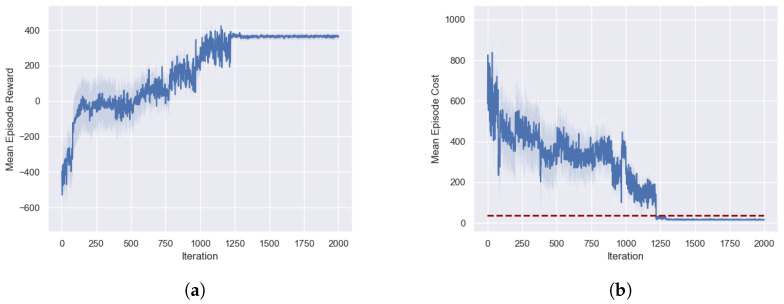
Analysis of performance during the training process of the SFRL-ACC strategy, including reward and safe values. (**a**) shows the score curve; (**b**) shows the cost curve, where the blue line represents the score and the red line represents the cost threshold.

**Figure 5 sensors-24-02657-f005:**
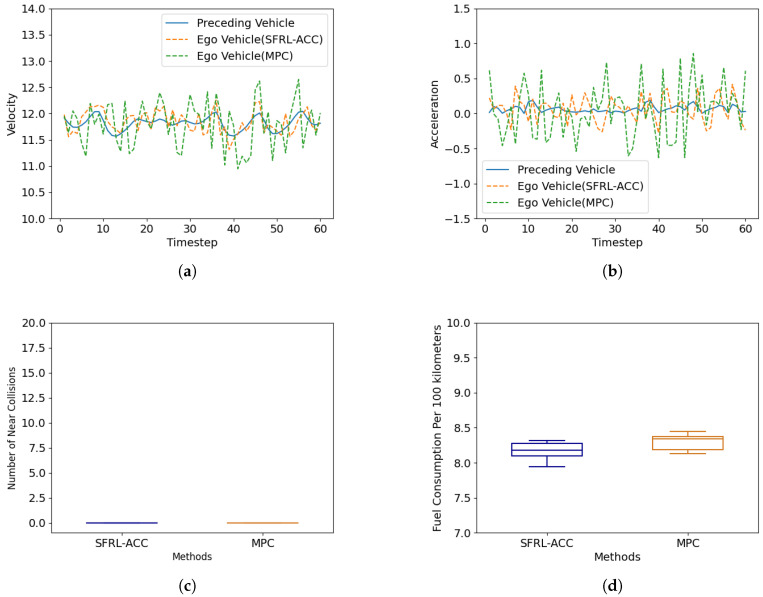
Performance comparison of the SFRL-ACC method and the MPC-based control method in terms of speed, acceleration, number of near collisions and fuel consumption per 100 km under the constant speed follow scenario. (**a**) shows the variation of speed with time steps, (**b**) shows the variation of acceleration with time, (**c**) shows the average number of collisions, and (**d**) shows the average white kilometer fuel consumption.

**Figure 6 sensors-24-02657-f006:**
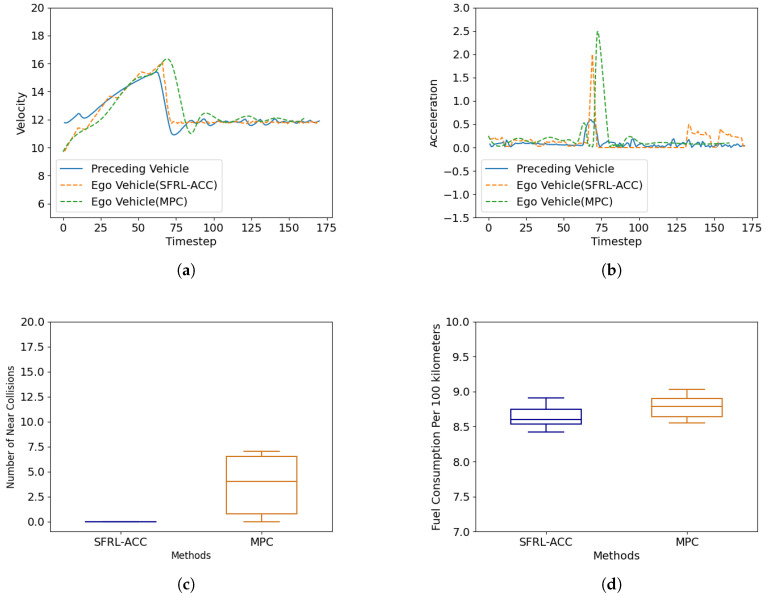
Performance comparison of the SFRL-ACC method and the MPC-based control method in terms of speed, acceleration, number of near collisions and fuel consumption per 100 km under the scenario of preceding vehicle braking. (**a**) shows the variation of speed with time steps, (**b**) shows the variation of acceleration with time, (**c**) shows the average number of collisions, and (**d**) shows the average white kilometer fuel consumption.

**Figure 7 sensors-24-02657-f007:**
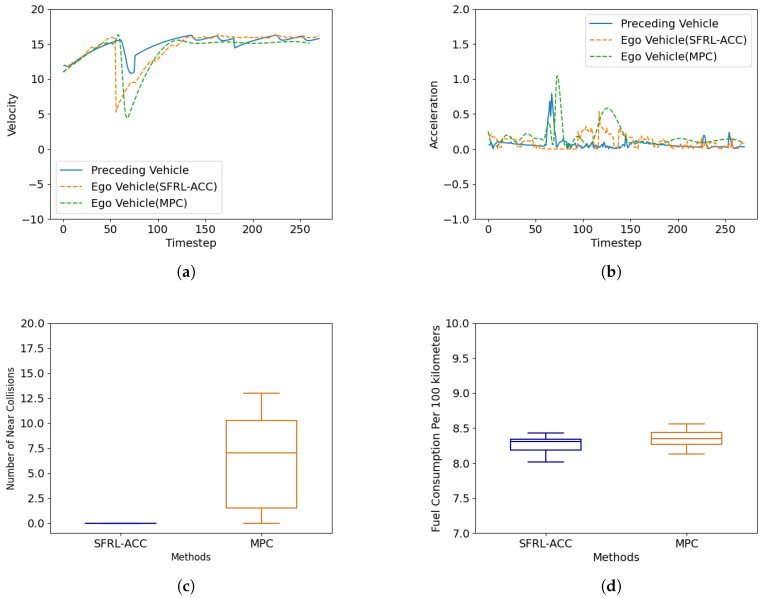
Performance comparison of the SFRL-ACC method and the MPC-based control method in terms of speed, acceleration, number of near collisions and fuel consumption per 100 km during cut-in scenarios. (**a**) shows the variation of speed with time steps, (**b**) shows the variation of acceleration with time, (**c**) shows the average number of collisions, and (**d**) shows the average white kilometer fuel consumption.

**Table 1 sensors-24-02657-t001:** Experimental parameter settings.

Parameters	Value
**CARLA Simulator**	-
Time step	0.1 s
Road width and lane width	14 m, 3.5 m
**SFRL-ACC**	-
Discount factor γ	0.99
Learning rate	1 × $10^{-3}$ → 0 (linearly)
Max KL divergence δ	0.001
Damping coefficient	0.01
Time steps $N_{t}$	500
Epoch $N_{e}$	2000
Cost limit	1
Hidden layer number	2
Hidden layer units	128
Policy std $\sigma_{0}$	1 → 0 (exponentially)
Policy std decrease index β	1.5 × $10^{-6}$
Coefficient of std ζ	1
Optimizer	Adam
TTC safety threshold $TTC_{s}$	4 s
**MPC**	-
Predictive horizon *T*	5
Velocity range	[0 m/s, 25 m/s]
Expect velocity $v_{\exp}$	16 m/s
Risk parameter α	0.005

## Data Availability

The data presented in this study are available upon request from the corresponding author.

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
