# Peer review of "Adaptive Cruise Control Based on Safe Deep Reinforcement Learning"

_sensors, 2024, doi:10.3390/s24082657_

Round 1
Reviewer 1 Report
Comments and Suggestions for Authors
Dear authors,
Please consider below comment to improve the paper.
1) Check again overall paper regarding grammatical errors. For example autono-mously, optimi-zation, computa-tion, dis-tance, cur-rently and etc.
2) Relocate Figure 1, after the paragraph of section 2.1.
3) Is it really experimental method to test SFRL-ACC algorithm ? Likely using Carla Simulator to test the method, thus simulation method.
4) If possible use plain background of Figure 5,6 and 7. Then, please increase the font clarity of these figures. Moreover, better to adjust the y-axis min and max region, so that the results can be clearly shown.
Reviewer 2 Report
Comments and Suggestions for Authors
This paper introduces SFRL-ACC, an adaptive cruise system utilizing Deep Reinforcement Learning (DRL) to address challenges faced by traditional ACC methods. It converts the ACC problem into a safe DRL framework called CMDP, then employs the PCPO-based SFRL-ACC algorithm tailored for CMDP. PCPO integrates safety constraints using Kullback-Leibler (KL) divergence, ensuring performance optimization while maintaining safety. Experimental results demonstrate SFRL-ACC's superiority in computation time, traffic efficiency, ride comfort, and safety over MPC-based ACC methods.
This paper is overall well-composed with some interesting ideas presented. Targeting for publication, some comments raised below need to be addressed:
1. The introduction should include some more recently published articles to better reflect the state of the art of the ACC (or vehicle longitudinal motion control). For example, authors can consider citing: [1] 10.1109/TIE.2023.3239878; [2] 10.1109/TMECH.2022.3146727
2. Although a control strategy is presented, its stability and robustness are not analyzed nor demonstrated.
3. There are if, else if, else logic inside the control algorithm. Will the switching between logics induce any oscillations? In practical applications, relay logics are employed to minimize frequent switching between conditions whenever switching logic is involved.
4. The energy consumption should be a performance metric to be presented and compared for the simulation study.
5. For transparency, the tuning for the MPC (benchmark solution) should be clarified.
Round 2
Reviewer 2 Report
Comments and Suggestions for Authors
The revision looks solid and the paper can be accepted in its current form.